

# Life on the edge—a changing genetic landscape within an iconic American pika metapopulation over the last half century

Kelly B. Klingler[1], Lyle B. Nichols[2], Evon R. Hekkala[3], Joseph A. E. Stewart[4] and Mary M. Peacock[5]

[1] Department of Environmental Conservation, University of Massachusetts Amherst, Amherst, Massachusetts, United States
[2] Department of Life Sciences, Santa Monica College, Santa Monica, California, United States
[3] Department of Biological Sciences, Fordham University, Bronx, New York, United States
[4] Department of Plant Sciences, University of California, Davis, Davis, California, United States
[5] Department of Biology, University of Nevada, Reno, Reno, Nevada, United States

Corresponding author
Mary M. Peacock,
mpeacock@unr.edu

## ABSTRACT

Declines and extirpations of American pika (*Ochotona princeps*) populations at historically occupied sites started being documented in the literature during the early 2000s. Commensurate with global climate change, many of these losses at peripheral and lower elevation sites have been associated with changes in ambient air temperature and precipitation regimes. Here, we report on a decline in available genetic resources for an iconic American pika metapopulation, located at the southwestern edge of the species distribution in the Bodie Hills of eastern California, USA. Composed of highly fragmented habitat created by hard rock mining, the ore dumps at this site were likely colonized by pikas around the end of the 19th century from nearby natural talus outcrops. Genetic data extracted from both contemporary samples and archived natural history collections allowed us to track population and patch-level genetic diversity for Bodie pikas across three distinct sampling points during the last half- century (1948–1949, 1988–1991, 2013–2015). Reductions in within-population allelic diversity and expected heterozygosity were observed across the full time period. More extensive sampling of extant patches during the 1988–1991 and 2013–2015 periods revealed an increase in population structure and a reduction in effective population size. Furthermore, census records from the last 51 years as well as archived museum samples collected in 1947 from a nearby pika population in the Wassuk range (Nevada, USA) provide further support of the increasing isolation and genetic coalescence occurring in this region. This study highlights the importance of museum samples and long-term monitoring in contextualizing our understanding of population viability.

## INTRODUCTION

The spatial distribution of an organism's habitat can vary at local, regional, and global scales (*Shen et al., 2013*; *Anderson et al., 2016*; *Klingler et al., 2021*). How an organism

navigates the spatial structure of its habitat will depend upon its body size, mode of locomotion, and life history (*Peacock & Smith, 1997*; *Pinto & MacDougall, 2010*; *Wauters et al., 2010*; *Neam & Lacher, 2018*). Climate change is likely to alter not only the extent of suitable habitat through changes to microclimate characteristics, but also the spatial organization of the remaining occupiable habitat for many organisms. As some habitat patches become unsuitable, the distance among habitable patches may increase, potentially impeding an organism's ability to find and move among otherwise suitable habitat areas. Such a dynamic may spiral into an extinction vortex (*Benson et al., 2019*) as populations or subpopulations become increasingly isolated and subject to demographic and genetic stochasticity (*Lande, 1993*), thereby reducing both persistence probability and evolutionary potential (*Blomqvist et al., 2010*).

Here, we examine the temporal dynamics of patch occupancy and the maintenance of genetic diversity from mid-20th to early 21st century for a unique metapopulation of the American pika (*Ochotona princeps*), a small, thermally sensitive alpine lagomorph (*Beever et al., 2010*; *Calkins et al., 2012*; *Stewart et al., 2015*; *Schwalm et al., 2016*). *O. princeps* is broadly distributed across mountain ranges of the western United States and Canada (*Smith & Weston, 1990*; *Hafner & Smith, 2010*) and is known to occupy a diversity of rocky habitats including natural talus, lava flows, and even hard-rock mining ore dumps (Fig. 1; *Severaid, 1955*; *Peacock, 1997*; *Peacock & Smith, 1997*; *Rodhouse et al., 2010*). Such habitat is naturally fragmented and varies widely in both patch size and spatial proximity which causes varying levels of connectivity across the range of this species. Because pikas already sustain a high resting body temperature (40.1 °C) very close to their upper lethal temperature (43.1 °C), the thermal consistency of the interstitial spaces of rocky talus habitat allows them to adjust their internal temperature through behavioral thermoregulation (*MacArthur & Wang, 1973*; *Wilkening, Ray & Varner, 2015*). This physiological sensitivity combined with a limited dispersal capacity (*Smith & Ivins, 1983*; *Peacock, 1997*) increases pika vulnerability to changing environmental conditions occurring under anthropogenic climate change (*Stewart, Wright & Heckman, 2017*).

Over the past several decades, pika populations have been disappearing, particularly in the mountain ranges of the Great Basin physiographic region, southern Utah, and lower elevation sites on the eastern slopes of the Sierra Nevada of California (*Beever, Brussard & Berger, 2003*; *Beever et al., 2010*; *Beever, Ray & Wilkening, 2011*; *Beever et al., 2016*; *Stewart & Wright, 2012*; *Nichols, Klingler & Peacock, 2016*; *Stewart et al., 2015*; *Stewart, Wright & Heckman, 2017*). These declines disproportionately affect the subspecies *O. princeps schisticeps*, whose distribution includes the Sierra Nevada and Great Basin mountain ranges of western and central Nevada (*Galbreath, Hafner & Zamudio, 2009*; *Galbreath et al., 2010*). In addition to evidence of shifts in the elevational distribution of pikas as a result of prehistoric climate change (late Pleistocene (Wisconsinan) to early Holocene; *Hafner, 1994*; *Grayson, 2005*), temperature metrics measured during 20th and 21st century human mediated climate change have repeatedly been identified as predictors of extant pika distributional change and persistence (*Beever, Brussard & Berger, 2003*; *Beever et al., 2010*; *Calkins et al., 2012*; *Stewart et al., 2015*; *Schwalm et al., 2016*). Average summer temperature and talus extent within a one km radius were predictors of both current

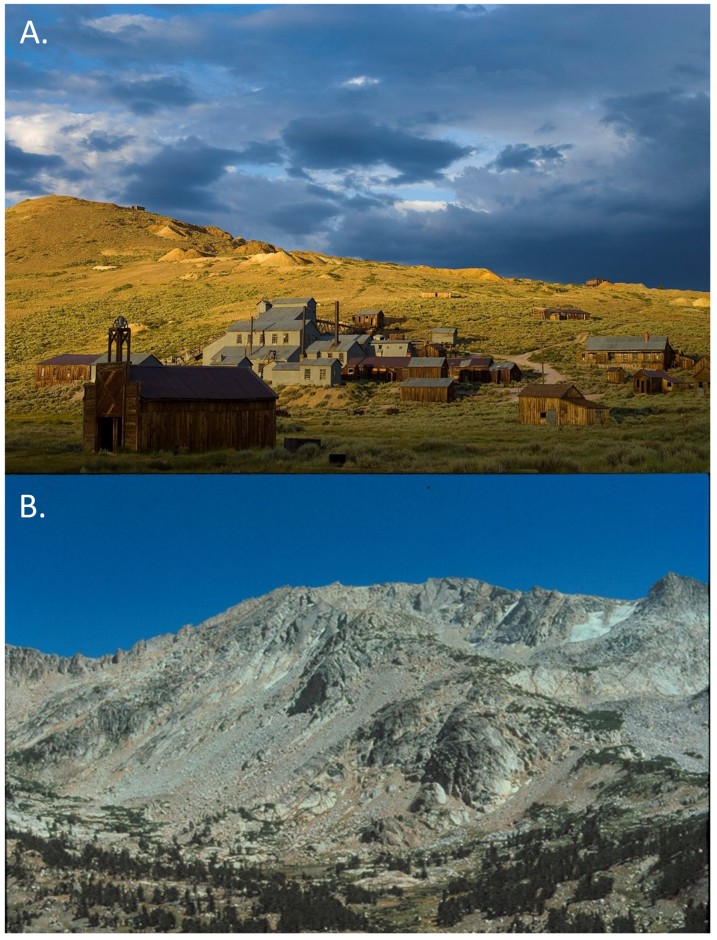

**Figure 1 Photographs of contrasting pika habitat.** (A) Bodie Historic State Park (photograph credit, Lyle Nichols), (B) Harvey Monroe Natural Area, Sierra Nevada (photograph credit, Mary Peacock).

occupancy by pikas, including the Bodie metapopulation studied here, as well as extirpation for historically occupied sites in the Sierra Nevada (*Stewart et al., 2015*).

However, at smaller spatial scales, non-climatic factors such as habitat configuration, connectivity, and suitable dispersal habitat are also likely to contribute to site-specific persistence probabilities, which likely vary across the species range (*Peacock & Smith, 1997*; *Jeffress et al., 2013*; *Castillo et al., 2014*; *Schwalm et al., 2016*). Indeed, pika populations continue to persist in some low-elevation sites (*Beever et al., 2008*; *Rodhouse et al., 2010*; *Manning & Hagar, 2011*; *Jeffress, Van Gunst & Millar, 2017*), which may reflect the nuances of local habitat suitability and potential for behavioral plasticity in this species (*Erb, Ray & Guralnick, 2011*; *Varner & Dearing, 2014*; *Moyer-Horner et al., 2015*; *Mathewson et al., 2017*).

The Bodie pika metapopulation is embedded within the Great Basin sagebrush-scrub plant community and habitat patches are composed of various sized ore dumps created by hard-rock mining during the late 19th and early 20th centuries. As a low elevational spur of the main Sierra Nevada mountain range, pikas are believed to have colonized these ore

dumps from neighboring natural talus patches sometime after the commencement of mining in the 1870s (*Severaid, 1955*). Since 1962, the remnants of the town and adjacent mining site containing the pika ore dump habitat have been protected by the State of California as Bodie State Historic Park (hereafter BSHP) (elevation, ~2,500 m). Recent surveys within the Bodie Hills indicate that while the BSHP population persists, most of the remaining natural rock outcroppings are currently unoccupied by pikas despite evidence of recent 20[th] and 21[st] century occupation (*Nichols, 2011*; *Stewart et al., 2015*; *Nichols, Klingler & Peacock, 2016*).

The BSHP pika population was first recognized by *Severaid (1955)* and has been under continuous study since 1969 (Fig. 2; *Smith, 1974a*, *1974b*, *1978*; *Peacock & Smith, 1997*; *Moilanen, Smith & Hanski, 1998*; *Nichols, 2011*; *Stewart et al., 2015*; *Nichols, Klingler & Peacock, 2016*; *Klingler et al., 2021*). This population has been the subject of studies on metapopulation dynamics (*Peacock & Smith, 1997*; *Moilanen, Smith & Hanski, 1998*), population genetic structure (*Peacock & Smith, 1997*), species distribution modeling (*Stewart et al., 2015*), and most recently a comparison of genome-wide patterns of population genetic variation among multiple pika lineages and subspecies (*Klingler et al., 2021*). Using DNA fingerprinting analysis, *Peacock & Smith (1997)* identified the presence of a mainland island metapopulation dynamic at BSHP with a steppingstone dispersal pattern occurring among smaller habitat patches. Despite the significant spatial structure of the habitat, *Peacock & Smith (1997)* observed a fluid gene flow dynamic such that genetic similarity among individuals both within and among talus patches fluctuated among years and resulted in a dynamic pattern of shifting population genetic structure over time. Contemporary genetic samples collected from BSHP during 2013–2015 revealed that this population has maintained moderate levels of nucleotide diversity ($\pi = 0.00147$) compared to populations from across the range of the species including Montana and Colorado (Rocky Mountains), which had higher levels of standing variation ($\pi$ range: 0.0025–0.0027), and populations in the Ruby-Humboldt ranges of Nevada, which had the lowest $\pi$ levels of all populations examined ($\pi$ range: 0.0007–0.0009) (*Klingler et al., 2021*). Despite intermediate levels of nucleotide diversity, BSHP showed evidence of having undergone the largest contraction of all populations sampled for this study (Tajima's D = 0.811; *Klingler et al., 2021*).

For the present study, the main goal was to investigate further how these patterns of genetic diversity and structure have fluctuated in the BSHP population over time, and to determine whether the previously documented metapopulation dynamic (*Peacock & Smith, 1997*) has been impacted by changes in patch occupancy and habitat suitability, which may be due to in part to a changing climate and increasing isolation. Here we test hypotheses of recent population contraction, a disrupted metapopulation dynamic and increasing regional isolation using a combination of genetic and annual census data. To examine the relationship between patch occupancy and maintenance of local population genetic diversity, we examine 51 years of BSHP population census data collected from the 1970s to present (*Moilanen, Smith & Hanski, 1998*; *Nichols, this study*) as well as population genetic data collected at BSHP in 1988–1991 and again in 2013–2015 on the majority of extant patches. To provide a historical context of the population genetic

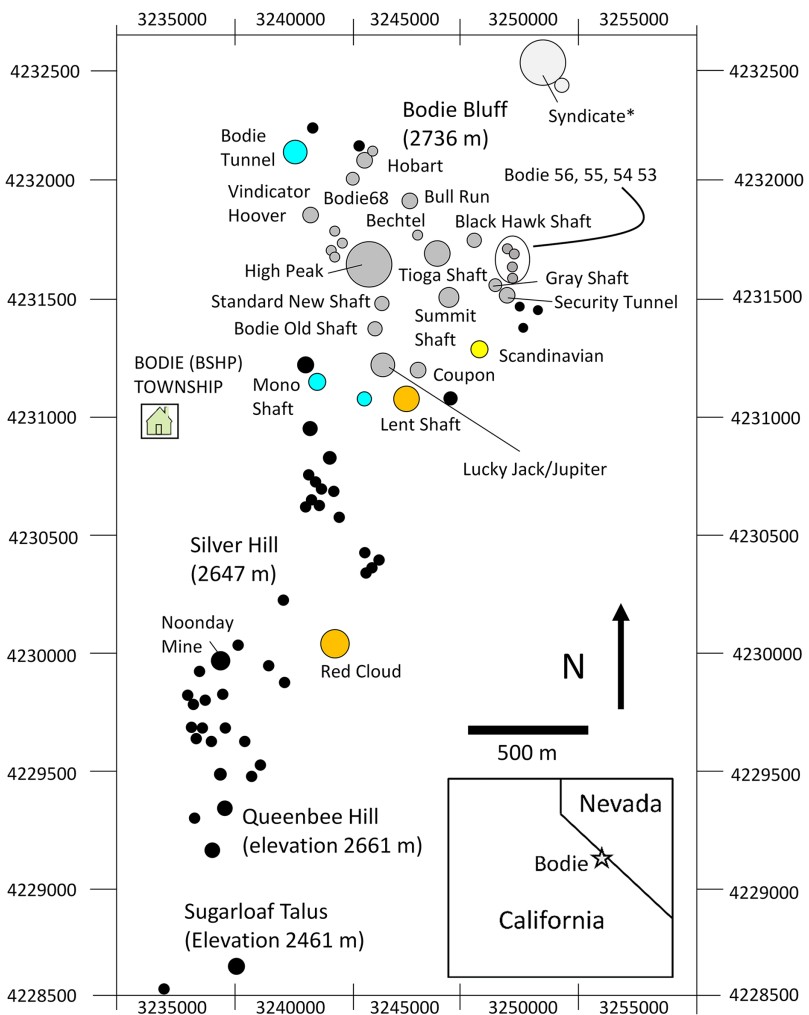

**Figure 2 Map study area talus patches.** Map of the study area showing orientation, area, and spatial configuration of BSHP ore dumps patches. Pikas were sampled for genetic analysis on a subset of the occupied patches over the three sampling periods as follows: aqua (1948–1949 only), yellow (1988–1991 only), orange (1949 and 2013–2015) and gray (1988–1991 and 2013–2015). Black filled circles represent unsampled patches that were largely unoccupied during the 1988–1991 and 2013–2015 survey periods. *The Syndicate was occupied and censused 1972–2018, however samples were not collected for genetic analysis from this patch.

diversity once present at this site, we also compare these two sets of more contemporary genetic samples (1988–1991 and 2013–2015) with specimens collected at BSHP in 1948 and 1949 by J.H. Severaid and curated at the Museum of Vertebrate Zoology (MVZ), University of California at Berkeley. We provide a comparison of population-level genetic diversity for each of the three time periods (1948–1949, 1988–1991 and 2013–2015) at BSHP with mid-20[th] century historical samples collected at another site in the region about 35 km northeast of BSHP (Big Indian Mountain (BIH), Wassuk Range, Mineral County, Nevada, collected by S.B. Benson and O.P. Pearson in 1947 and curated at MVZ).

This study is unique in being able to provide a long-term and fine-scale perspective on patterns of patch occupancy as well as levels of population genetic diversity and structure.

Recent reductions in population size and losses of genetic diversity (*Klingler et al., 2021*) may indicate a fundamental change in the metapopulation dynamic at this site. Evidence of a disrupted metapopulation dynamic (*Peacock & Smith, 1997*) and increasing regional isolation over the 20th and 21st centuries would include: (1) a reduction in the proportion of patches occupied over time, (2) evidence of genetic coalescence, where standing genetic variation is reduced to the amount that can be maintained in consistently occupied habitat (*Gilpin, 1991*), and genetic erosion *via* a decline in allelic richness as well as observed and expected heterozygosity, (3) a decrease in the effective population size (Ne), (4) increasing isolation among occupied ore dump habitat patches within the BSHP site and (5) increased genetic differentiation between the BSHP population over time with mid-century BIH.

## METHODS

### Ethical approval and consent to participate
This research was approved by the Institutional Animal Care and Use Committees (IACUC) at the University of Nevada, Reno (protocol No. 00557). Scientific collecting permits were obtained from the California Department of Parks and Recreation (No. 2014-03); California Department of Fish and Wildlife (SC-012184); US Department of Agriculture (USDA) Forest Service and USDA Forest Service, Inyo National Forest (No. LVD14021 and No. RMT143).

### Study species
Pikas are small alpine lagomorphs (~150 g), which inhabit rocky substrates (talus) in the mountain ranges of the American west. Adult pikas are individually territorial and defend their territories from conspecific intrusions (*Smith & Ivins, 1983*; *Peacock, 1997*). As lagomorphs, they do not hibernate, and store vegetation collected during the summer months in one or two centrally placed "haypiles" on their territories that they subsequently feed on during the winter months (*Smith & Weston, 1990*). Haypile sites, which define the effective center of an adult's home range, are maintained in the same location even when territory ownership changes (*Peacock, 1997*). Territories are typically found at the talus-vegetation interface where there is easy access to forage and escape cover from predators. Suitable locations for territories on a talus slope are limited and the number of territories found in any one location is relatively static. Juveniles are born in late spring/early summer and remain on their natal territory until they are weaned at which time they begin exploratory movements to locate open territories (*Smith & Ivins, 1983*). Territory ownership is essential for overwinter survival and juveniles typically colonize the first available territory they encounter during their natal summer (*Peacock, 1997*; *Peacock & Smith, 1997*).

### Climate
To examine trends in summer temperature at the study site from 1940 to present day, we used monthly historical PRISM data (*Daly, Neilson & Phillips, 1994*) extracted, using bilinear interpolation, at the mean location of the 82 habitat patches surveyed in annual censuses. Mean summer temperatures were calculated as the means of daily high and low
temperatures for June, July, and August of each year. We used linear regression to assess trend over times and a two-sided Welch Two sample t-test to compare temperatures from 1940–1960 with temperature from 2000–2030. Analyses were conducted in R version 4.3 (*R Core Team, 2023*).

## BSHP population census surveys

Eighty-two individual pika habitat patches, representing the majority of the total suitable habitat at BSHP (~100 patches) were surveyed in 1972, 1977, 1989, 1991–2001, 2003–2006, and 2008–2020 (*Smith, 1974a*; *Moilanen, Smith & Hanski, 1998*; Nichols, this study). The presence of fresh fecal pellets and/or active haypiles within each pika territory was used to determine whether each patch was occupied and to estimate the number of pikas occupying each patch (*Dearing, 1997*; *Moilanen, Smith & Hanski, 1998*). While all 82 patches were surveyed during later survey years (*i.e.*, after 2008), earlier years have some idiosyncratically distributed missing data (*i.e.*, ~2% missing data, overall). To ameliorate the influence of missing data, we linearly imputed missing occupancy and count data over time (Fig. S1). We used nonparametric tests (*i.e.*, two-sided Spearman's rank correlation) to evaluate the strength and significance of trends over time for patch occupancy, population count, and number of pikas per occupied patch. We used quasibinomial logistic and negative binomial regressions, respectively for occupancy and population count data, to visualize trends over time. Population count trends were evaluated for the entire patch network as well as the southern (all patches south of Silver Hill), middle (patches on Silver Hill south of Mono Shaft) and northern (all patches north of and including Mono Shaft) patch networks (see Fig. 2). Patch occupancy and number of pikas per occupied patch were evaluated for both the full period of record (1972–2022) and for the subset of years that excludes the first two survey years (1989–2022). Analyses were conducted in R version 4.3 (*R Core Team, 2023*).

## Genetic sample collection

**Mid-20[th] century museum samples (1947–1949).** Approximately 200 pika museum specimens collected during the mid-20[th] century at BSHP by *Severaid (1955)* are curated in the mammal collections in the Museum of Vertebrate Zoology, University of California at Berkeley. Additional museum preserved samples are available from a number of historically occupied pika sites across Nevada. K. Klingler was given access to 16 specimens from BSHP collected by Severaid 1948–1951 (Supplemental Material, Table S1). E. Hekkala was given access to an additional 19 samples collected from BHSP 1948–1949 by Severaid and Big Indian Mountain, Wassuk Range, Nevada (BIH) collected by S.B. Benson and O.P. Pearson in 1947 (Table S1). A 2–5 mm square piece of dried museum pelt was cut from the ventral lip and placed in a 1.5 ml locking tube. We included the BIH historical samples to test for a pattern of increasing regional isolation of the BSHP population over the time frame from 1948–2015.

**Late 20[th] century samples (1988–1991).** Pika ear tissue was collected from live-trapped individuals in BSHP as part of a metapopulation dynamics study of 24 occupied patches in the northern portion of the study area (1988–1990; *Peacock & Smith, 1997*). After

completion of the study, the remaining DNA samples were stored in a −20 °C freezer until present ($n = 114$). All adult samples ($n = 75$) from the 1988–1991 sampling period were used to estimate population level genetic metrics for this study.

**Early 21st century samples (2013–2015).** Fecal DNA samples were collected from 26 occupied habitat patches in the northern half of the BSHP site during June-September 2013–2015 to be used for comparison to the earlier time periods. Adult pikas defecate in well-defined latrine sites within their territories (*Nichols, 2011*). Differences in size between adult and juvenile animals make identification of adult *vs* juvenile fecal pellets unambiguous. Multiple fresh fecal pellets (~6–8) from single latrine sites identified using methods of *Nichols (2011)*, were collected per individual using stainless steel serrated tipped forceps. The forceps were wiped down between samples, submerged in 70% to 100% ethanol and allowed to air dry between sample collections to remove cellular debris. Two to five fecal pellets collected per individual animal were immediately transferred to a 2.0 ml vial with 500 µl of ATL tissue lysis buffer (DNeasy96 Blood and Tissue kits; QIAGEN, Hilden, Germany) after collection in the field. At the end of each collection period (~2–4 h), 50 µl of ice-cold proteinase K was added to 400 µl of supernatant from each sample for storage until DNA extraction. We did not homogenize the pellets, but inverted them multiple times once placed in the lysis buffer in order to wash the epithelial cells from the pellet surface prior to pipetting the supernatant and adding the proteinase K. In order to rule out multiple individuals in DNA isolated from multiple fecal pellets and to also control for possible contamination during pellet collection, we did not score any questionable low amplitude alleles and eliminated individuals that had three or more scorable alleles at any locus from the analysis. We also tested for both heterozygote excess and deficiency per locus per patch subpopulation for the fecal DNA samples using the Fisher Exact test in GENEPOP (version 4.2; *Rousset, 2008*).

## DNA isolation and polymerase chain reaction (PCR) amplification

Total genomic DNA was isolated according to the manufacturer's protocol (DNeasy96 Blood and Tissue kits; QIAGEN, Hilden, Germany) with modifications specific to each sample type. Details regarding modifications to DNA extraction protocol for fecal and museum genetic samples can be found in *Hekkala et al. (2011)* and *Peacock et al. (2017)*. All DNA extractions and polymerase chain reactions (PCR) for mid-20th century museum samples (1947–1949) were conducted in a separate laboratory in order to avoid DNA contamination from contemporary pika samples that have been isolated in our laboratory.

We used eight di-, tri-, and tetranucleotide repeat, polymorphic microsatellite loci of the 28 that have been developed for the American pika (OCP4, OCP6, OCP7, OCP8, OCP9, *Peacock, Kirchoff & Merideth, 2002*; OCP10, OCP17, OCP21; GQ461705.1–GQ461722.1, direct submission Genbank). We chose microsatellite loci that were smaller (in terms of fragment length) and that produced a consistent and scorable PCR product from both the fecal and museum DNA samples. PCRs were carried out in 12 µl reaction volumes on a MBS Satellite 0.2 G thermal cycler (Thermo Fisher Scientific, Waltham, MA, USA), and included a 15 min hot start at 94 °C, followed by 41 cycles at 94 °C, 30 s each for denaturing, a touch down annealing temperature for 90 s (seven cycles of 65 °C, followed
by seven cycles of 61 °C, and seven cycles of 58 °C each were completed with a final 20 cycles at 55 °C) and an elongation step at 72 °C for 30 s (modified from *Peacock, Kirchoff & Merideth, 2002*). The first 21 cycles amplified the locus specific primer and the final 20 cycles added a fluorescently labeled M13 tail to the PCR product. All PCR products were diluted to an appropriate concentration for use with an automated sequencer and 1 µl of PCR product was added to 19 µl of HiDye Formamide/LIZ500 size standard for fragment analysis (Applied Biosystems, Waltham, MA, USA). Fragment size analysis was carried out on an Applied Biosystems 3730 DNA Genetic Analyzer at the UNR Nevada Genomics Center (https://www.unr.edu/genomics/services). All alleles were scored, binned, and genotyped using ABI Prism GeneMapper software (version 5.0).

## Allelic dropout and genotyping error rates for fecal and museum samples

DNA from low quality fecal and preserved museum samples can lead to genotyping errors, which may bias estimates of population genetic metrics (*Creel et al., 2003*; *McKelvey & Schwartz, 2005*). Common genotyping errors include allelic dropout, preferential allele amplification, and false alleles. For contemporary fecal samples, each individual was genotyped multiple times (2–4) at all loci using separate PCR reactions until a consensus genotype was reached. Individuals were only included in genetic analyses if a consensus genotype was achieved through repeated, successful amplification. For museum samples, DNA was amplified in five separate PCR reactions for all loci and individuals were included in genetic analyses only if a consensus genotype was reached at each locus for three of the five PCR reactions.

Once consensus genotypes were obtained, we identified and removed duplicate samples from the 2013–2015 fecal DNA dataset using the Multilocus Matches DS module in GenAlEx (version 6.501; *Peakall & Smouse, 2006*, *2012*). In addition, we used the probability-of-identify function in GenAlEx for the 2013–2015 data to assess our power to detect individuals using the eight microsatellite loci dataset. As the 1988–1991 data were tissue samples obtained from individuals who were individually marked with ear tags, individual identity was not an issue.

Locus-specific error rates for allelic dropout ($\epsilon_1$), false alleles ($\epsilon_2$), and genotyping success rate were estimated using replicate genotypes for fecal DNA and museum specimens through a maximum-likelihood based approach (program Pedant, Delphi version 7.0) (*Johnson & Haydon, 2007*). In addition, the programs MicroChecker (version 2.2.3; *Van Oosterhout & Hutchinson, 2004*) and FreeNA (*Chapuis & Estoup, 2007*) were used to test for the presence of null alleles, and genotyping errors. If a locus exhibited systematic patterns of deviation from HWE, showed evidence of null alleles or allelic dropout it was removed from the analysis.

## Population genetic analyses

We used FSTAT (version 2.9.3.2; *Goudet, Perrin & Waser, 2002*) to quantify the number of alleles (Na), to test for linkage disequilibrium and deviations from Hardy-Weinberg equilibrium ($F_{IS}$) per locus per time period and to calculate a global θ for each time period

(1948–1949, 1988–1991, 2013–2015). We also used FSTAT to calculate pairwise $F_{ST}$ estimates for individual patches sampled in both 1988–1991 and 2013–2015 that had sufficient data for comparisons (patch occupancy $n \geq 3$ individuals). We also calculated a pairwise $F_{ST}$ at the BSHP population scale for comparison between the two time periods (1988–1991 and 2013–2015) that had data from extensive sampling.

In order to correct for differences in sample size between the three sampling periods, a hierarchical rarefaction approach was used in the program HP-RARE (version 1.0; *Kalinowski, 2005*) to calculate estimates of allelic richness (Rs) and private alleles (Pa). The average observed (Ho) and expected (He) heterozygosity per locus per sampling period were calculated using Microsatellite Toolkit in Excel (*Park, 2001*). Differences in Rs, Pa, Ho and He among time periods were tested with the time period as a categorical variable using ANOVA in SYSTAT (version 8.0, 1998) together with *post hoc* Tukey tests. Due to the small sample size for 1948–1949, effective population size (Ne) was estimated for the 1988–1991 and 2013–2015 time periods only using the linkage disequilibrium module based on a single point sample in NeEstimator (version 1.3; *Peel, Ovenden & Peel, 2004*; *Waples & Do, 2010*). We conducted an AMOVA in GenAlEx for the 1988–1991 and 2013–2105 population wide datasets to assess the partitioning of genetic variation within and among habitat patches.

A Principal Coordinates Analysis (PCoA) in GenAlEX, which uses a pairwise genetic distance matrix calculated from multi-allelic, multi-locus microsatellite data, was used to characterize any changes in the spatial partitioning of genetic diversity within BSHP across the three time periods and between BSHP and BIH. We then used a Bayesian genotype clustering approach (STRUCTURE version 2.3.4; *Pritchard, Stephens & Donnelly, 2000*) to assess differences between genotype frequencies and therefore assignment probability to distinct genotype clusters ($k$) across the three time periods using a 50,000-iteration burn-in followed by ten 500,000 Markov Chain Monte Carlo (MCMC) replications per $k$, for $k = 1$–10 and the $\Delta k$ method to determine the optimal number of genotype clusters (*Evanno, Regnaut & Goudet, 2005*).

Results from an earlier study which tracked tagged individuals and used variable number of tandem repeat minisatellite nuclear DNA data (*Peacock & Smith, 1997*), revealed frequent dispersal among patches together with high within patch mortality that resulted in a fluid movement dynamic among patches and rapidly changing population genetic structure across years. To assess changes in movement dynamics and population genetic structure we compared the patch level microsatellite datasets for 1988–1991 and 2013–2015 using STRUCTURE (version 2.3.4, *Pritchard, Stephens & Donnelly, 2000*). To further characterize genetic changes within and among patches between the two time periods, 1988–1990 and 2013–2015, we calculated population level relatedness ($r$) for each time period using the *Queller & Goodnight (1989)* method in GenAlEx, as well as $r$ per patch for all patches with $n > 3$ individuals. We also calculated a genetic similarity metric among all individuals across the entire population, among individuals per patch and made pairwise comparisons of genetic similarity between patches sampled in 1988–1991 and again in 2013–2015 for which we had sufficient data (patch occupancy $n \geq 3$ individuals) using the allele sharing matrix module in Microsatellite toolkit in Excel (*Park, 2001*).

We then conducted paired t-tests for matched patch pairs for those patches with sufficient data from both 1988–1990 and 2013–2015 to test for statistically significant changes in genetic similarity within patches over time.

## RESULTS

### Climate

From 1940 to 2022, mean summer temperature increased by 0.38 °C/decade (95% CI [0.27–0.50], $P = 4.7 \times 10^{-9}$) (Fig. 3). Mean summer temperatures from 2000–2020 were 2.07 °C (95% CI [1.51–2.63]) higher than mean summer temperatures from 1940–1960 ($P = 9.1 \times 10^{-9}$).

### Patch occupancy

Across the entire patch network, amidst interannual fluctuations, the percentage of patches occupied has declined over time (1972–2022: $r_s = -0.47$, $P = 0.0062$; 1989–2022: $r_s = -0.36$, $P = 0.049$; Fig. 4A) while the population count has remained relatively unchanged ($r_s = -0.15$, $P = 0.42$; Fig. 4C), resulting in an increased concentration of pikas on remaining occupied patches (1972–2022: $r_s = 0.45$, $P = 0.0092$; 1989–2022: $r_s = 0.39$, $P = 0.032$; Fig. 4D). Data from census surveys show the collapse of the middle and southern pika subpopulations occurring sometime during the 1980s followed by intermittent extirpations and recolonizations from 2008–2022. Population counts in these regions declined significantly over time (middle: $r_s = -0.67$, $P = 0.000017$; southern: $r_s = -0.52$, $P = 0.0021$). The southern and middle portions of the study area are dominated by small (1–4 territories) and medium (5–9 territories) patches (Fig. 4B) and have never been fully reoccupied and remain largely vacant (Fig. 4C). After collapse in the 1980s, subpopulations in the southern patch network (all patches south of Silver Hill, see Fig. 2) experienced a small increase in the number of patches occupied in 2003 and 2004 (4–5 patches occupied), but occupancy declined to one patch in 2005 before complete extirpation by 2008 (Fig. 4C). Since then, there have been a few recolonization attempts in the southern patch network, but no patch has remained consistently occupied; from 2008–2022 three patches have experienced intermittent occupancy, with up to two patches occupied at a time. The talus patches in the middle patch network (patches on Silver Hill south of Mono Shaft, see Fig. 2) were all mostly unoccupied by the early 2000s and have remained so (Fig. 4C). In contrast, the northern patch network (all patches north of and including Mono Shaft) has maintained consistently occupied patches over the same period and although population size has fluctuated on these patches, the northern network has not experienced a population decline ($r_s = 0.06$, $P = 0.75$; Fig 4C).

### Allelic dropout, null alleles and genotyping error rates

Null alleles were probable at multiple loci in 1988–1991 (OCP10 and 21) and 2013–2015 (OCP10 and 17), but there were no consistent patterns across loci, patch network, or time. The fecal DNA samples had an amplification success rate of 95.5% with average rates of allelic dropout and false alleles at 2.9% and 1.6%, respectively (Table 1). The museum specimens had a lower amplification success rate (90.2%), higher allelic dropout (3.9%) and more false alleles

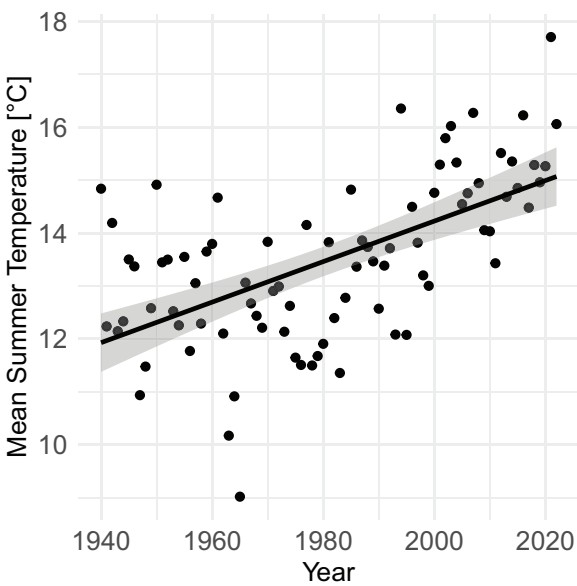

**Figure 3** **Increase in mean summer temperature (June, July, August) from 1940–2022 (PRISM data).**
Summer temperature increased by 0.38 °C/decade (linear regression, $P = 4.7 \times 10^{-9}$).

(4.4%) compared to fecal DNA samples from extant animals, which was expected considering age and degradation of samples. The ear tissue samples collected during 1988–1990 were not repeatedly amplified due to the high-quality nature of these samples.

## Genetic diversity

A total of 298 unique adult individuals were successfully genotyped from the contemporary and historic BSHP mining district as well as the historical samples from the Big Indian Mountain (BIH) site (tissue samples 1947–1949, $n = 22$; 1988–1990, $n = 75$; fecal samples 2013–2015, $n = 191$). The 1947–1949 museum samples failed to amplify at OCP7, which was subsequently removed from all analyses that included museum samples. The 1948–1949 dataset was too small to calculate heterozygous excess and deficiency on a per patch basis. We did not observe heterozygous excess or deficiency for any locus on any per patch for the 1988–1991 dataset. However, we observed both heterozygous excess and deficiency in the 2013–2015 fecal DNA dataset, but primarily heterozygous excess, for subpopulations on five of the largest 26 patches sampled consistent with a signature of genetic bottleneck (Bull Run, $X^2 = 56.08$, df = 14; High Peak, $X^2 = 52.86$, df = 16; Hobart, $X^2 = 36.55$, df = 16; Tioga Shaft, $X^2 = 36.83$, df = 14; Vindicator Hoover, $X^2 = 49.77$, df = 14; $P = 0.0001$). Because fecal samples were collected fresh and individual collections were made from distinct latrine sites on individual territories, we conclude that we sampled single individuals per fecal sample collected. The probability of identity calculated for the 2013–2015 fecal DNA dataset after duplicate samples were removed using the Multilocus Matches DS module in GenAlEx was 1.9E-05.

We observed linkage disequilibrium at one locus pair (OCP10 × OCP21, $P = 0.0001$, 7,280 permutations) in the 1988–1991 dataset for the High Peak subpopulation only.

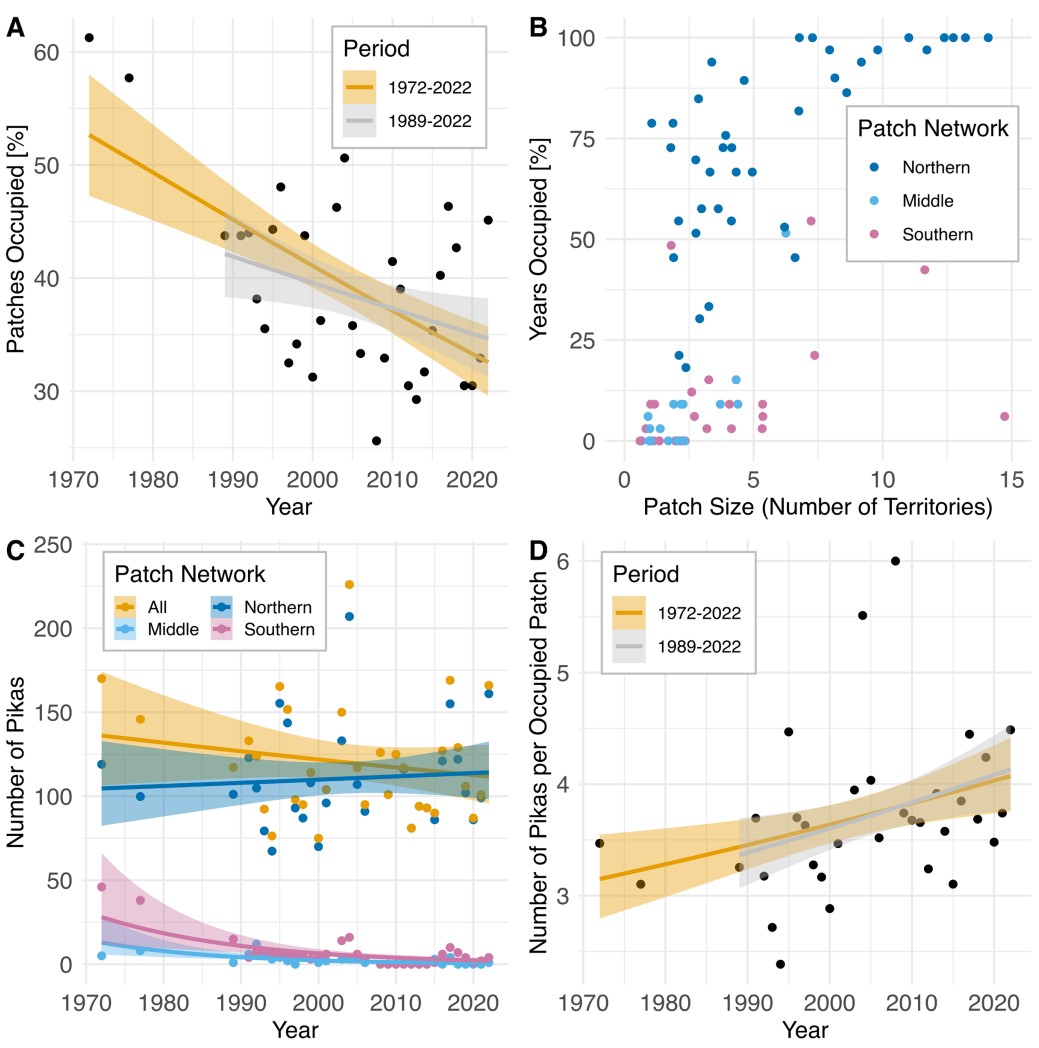

**Figure 4** **Pika population trajectories with 95% CIs and patch characteristics at Bodie State Historic Park based upon population survey data collected 1972–2022.** Non-parametric *P*-values were calculated using Spearman rank order correlation (two-sided test). (A) Decline in the proportion of habitat patches occupied over time (1972–2022: $r_s = -0.47$, $P = 0.0062$; 1989–2022: $r_s = -0.36$, $P = 0.049$). (B) Characteristics of 82 individual habitat patches surveyed. Points are jittered to reduce overlap. See text and Fig. 2 for delineation of northern, middle, and southern patch networks. (C) Amidst modest but nonsignificant change in population size in the northern patch network ($r_s = 0.06$, $P = 0.75$) and across all patches ($r_s = -0.15$, $P = 0.42$) the middle ($r_s = -0.67$, $P = 0.000017$) and southern ($r_s = -0.52$, $P = 0.0021$, $P = 0.005$) patch networks have experienced decline and intermittent extirpation. (D) Increase in the concentration of pikas on remaining occupied patches after extirpation of subpopulations in the middle and southern patch networks (1972–2022: $r_s = 0.45$, $P = 0.0092$; 1989–2022: $r_s = 0.39$, $P = 0.032$).

For the 2013–2015 dataset, three locus pairs were in linkage disequilibrium, but only for two patches, High Peak (OCP8 × OCP9) and Hobart (OCP4 × OCP8, OCP8 × OCP10) ($P = 0.00009$, 10,800 permutations). On a BSHP population wide basis, a significantly larger $F_{IS}$ was observed for OCP10 in the 1948–1949 and 1988–1991 datasets ($F_{IS} = 0.683$, 0.501, $P = 0.0018$, 560 randomizations) and significantly smaller $F_{IS}$ than random for OCP4 ($F_{IS} = -0.286$), OCP9 ($F_{IS} = -0.236$) and OCP 21 ($F_{IS} = -0.232$) ($P = 0.0018$, 560
**Table 1 Locus-specific rates of allelic dropout ($\varepsilon_1$) and false alleles ($\varepsilon_2$) for museum and fecal DNA samples.**

| | 1949 Museum sample DNA | | | 2015 Fecal DNA | | |
|---|---|---|---|---|---|---|
| Locus | $\varepsilon_1$ (Allelic dropout rate) | $\varepsilon_2$ (False allele rate) | % Success | $\varepsilon_1$ (Allelic dropout rate) | $\varepsilon_2$ (False allele rate) | % Success |
| OCP 4b | 0.031 | 0.138 | 0.923 | 0.000 | 0.010 | 0.930 |
| OCP 6b | 0.091 | 0.036 | 0.846 | 0.000 | 0.032 | 0.987 |
| OCP 7 | – | – | – | 0.095 | 0.000 | 0.917 |
| OCP 8 | 0.050 | 0.050 | 0.969 | 0.006 | 0.000 | 0.974 |
| OCP 9b | 0.000 | 0.015 | 0.892 | 0.006 | 0.000 | 0.939 |
| OCP 10 | 0.031 | 0.046 | 0.954 | 0.023 | 0.000 | 0.996 |
| OCP 17 | 0.015 | 0.000 | 0.877 | 0.052 | 0.070 | 0.956 |
| OCP 21 | 0.033 | 0.050 | 0.877 | 0.014 | 0.033 | 0.934 |
| Average | 0.035 | 0.039 | 0.909 | 0.034 | 0.015 | 0.956 |

**Note:**
Genotyping success rate per locus averaged across all samples (%).

randomizations) in the 2013–2015 dataset (Table S2). All genetic diversity metrics, number of alleles (Na), allelic richness (Rs), observed (Ho) and expected (He) heterozygosity and inbreeding coefficient ($F_{IS}$) are summarized per locus per time period in Table S2.

We observed significant declines in Rs ($F_{2,18} = 10.228$, df = 2, $P = 0.001$), and Pa ($F_{2,18} = 20.89$, $P = 0.000$), and He ($F_{2,18} = 8.774$, $P = 0.006$), but not Ho ($F_{2,18} = 0.087$, $P = 0.917$) over the three time periods at the BSHP site (1948–2015; Figs. 5A–5D ). *Post hoc* Tukey tests show that the 1948–1949 values for Rs and Pa were significantly higher than those for 1988–1991 and 2013–2015 ($P = 0.006$), whereas the 1988–1991 and 2013–2015 values did not differ ($P = 0.911$, 0.732 respectively). The 1948–1949 He estimate was significantly higher than 2013–1015, but not 1988–1991 ($P = 0.005$, 0.107 respectively), whereas the 1988–1991 and 2013–2015 He values did not differ ($P = 0.296$).

## Population genetic analyses

AMOVA results show 91% of the genetic variance was within individuals, 7% among individuals and 2% among patches in 1988–1991, with 91% within individuals and 9% among patches in 2013–2015 (999 permutations). Effective population size also declined between the two later time periods 1988–1991 (Ne = 76, 95% CI [47.4–149.8]) and 2013–2015 (Ne = 47.1, 95% CI [38.4–58.3]; Fig. 5E).

The small sample size from 1948–1949 precludes an accurate assessment of relatedness on a population wide basis for this early time period, but there was a clear trend in the increase in relatedness over time with low average relatedness in 1948–1949 ($r = -0.068$, 95% CI [−0.113 to 0.128]) and 1988–1991 ($r = 0.024$, 95% CI [−0.060 to 0.060]) but an $r$ value consistent with an average relatedness among first cousins ($r = 0.158$, 9% CI [−0.028 to 0.021]) in 2013–2015 (Fig. 5F). Consistent with the relatedness results, pairwise genetic similarity among all BSHP individuals per time period show a significant increase in population level genetic similarity across time (Kruskal-Wallis Test Statistic = 1,780.085, df = 2, $P = 0.000$; Fig. 6A). For the subset of patches for which we had sufficient data from

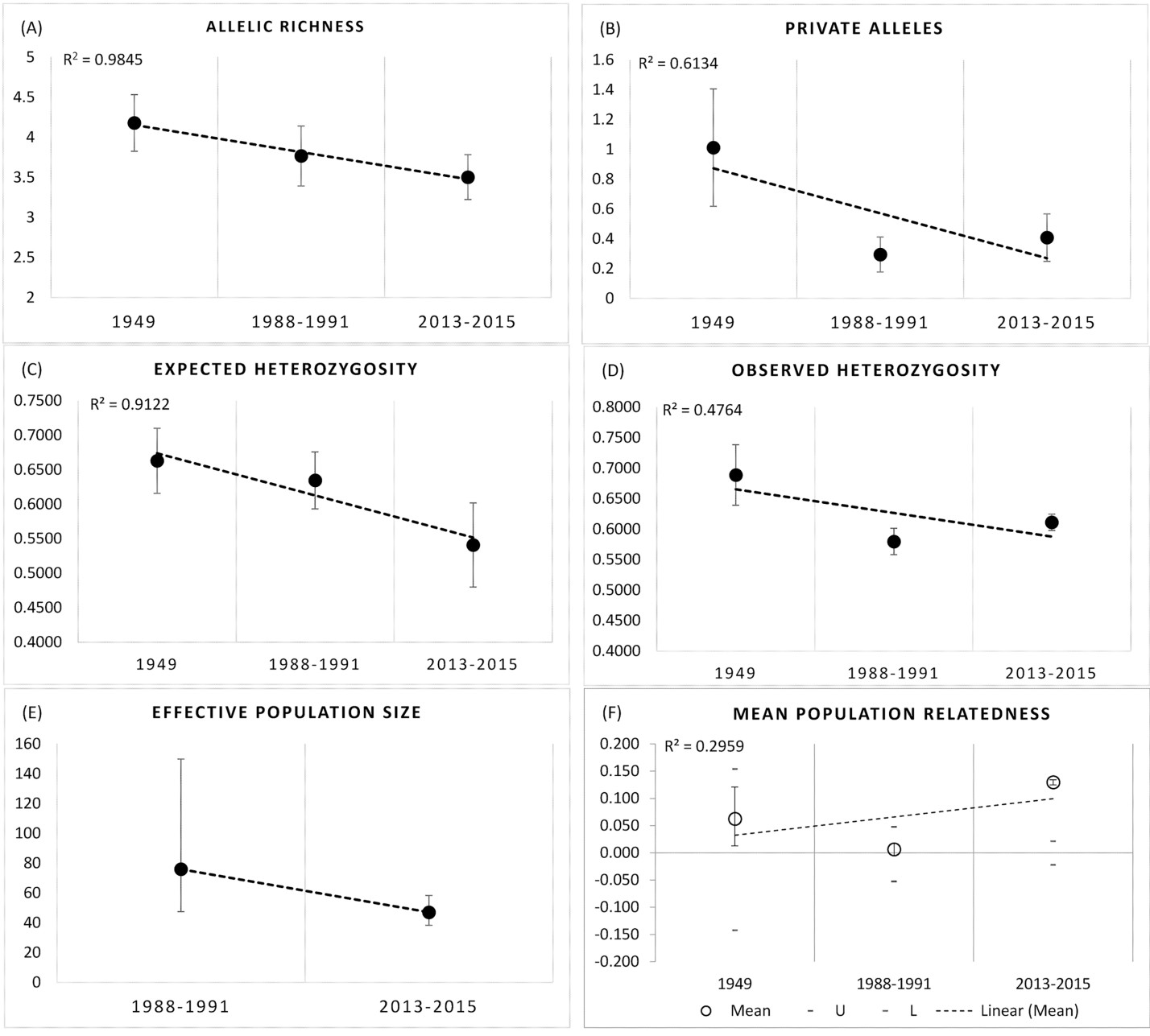

**Figure 5 Genetic metrics for entire BSHP population for each time period 1948–1949, 1988–1991, 2013–2015.** We observed significant declines in (A) allelic richness (Rs) ($F_{2,18} = 10.228$, df = 2, $P = 0.001$), (B) private alleles (Pa) ($F_{2,18} = 20.89$, $P = 0.000$), and (C) expected heterozygosity (He) ($F_{2,18} = 8.774$, $P = 0.006$), but not (D) observed heterozygosity (Ho) ($F_{2,18} = 0.087$, $P = 0.917$) over time. (E) Effective population size (Ne) declined from 1988–1991 to 2013–2015, and (F) average relatedness (*r*) for entire BSHP population per time period (1948–49, 1988–1991, 2013–2015) increased over time. The upper and lower error bars bound the 95% confidence interval about the mean relatedness values as determined by bootstrap resampling. Upper (U) and lower (L) confidence limits bound the 95% confidence interval about the null hypothesis of "No Difference" in relatedness across the populations as determined by permutation.

both 1988–1991 and 2013–2015, within patch comparisons among individuals also showed a significant increase in genetic similarity between time periods (Mann-Whitney U, 10.5–97595.5, df = 1, $P = 0.000$–0.009; Fig. 6B), as well as increases in average

relatedness for all patch pair comparisons across years with the exception of High Peak, which had an average $r$ of 0.01 and −0.003 for the 1988–1991 and 2013–2015 respectively (Fig. S2). We note that High Peak is centrally located within the northern patch network and also one of the largest patches present in the study area.

We did not observe significant pairwise $F_{ST}$ estimates among patch subpopulations for the 1988–1991 dataset, but did observe multiple significant $F_{ST}$ estimates for patch pairs during the 2013–2015 time period ($n$ = 12 significant $F_{ST}$ estimates, $P$ = 0.0006, 1,560 permutations; Table S3). The majority of the significant pairwise $F_{ST}$ estimates were between core and peripheral populations in the northern patch network (Table S3). Global estimates of θ for the latter two time periods, with extensive sampling across all occupied patches, show an increase in population genetic structure (1988–1991, θ = 0.021, 95% CI [0.012–0.03]; 2013–2015, θ = 0.087, 95% CI [0.067–0.107]). Pairwise $F_{ST}$ between BSHP and BIH were significant across all time periods, with increasing pairwise $F_{ST}$ over time (BIH 1947 and BSHP 1948–1949 $F_{ST}$ = 0.152, BIH 1947 and BSHP 1988–1991 $F_{ST}$ = 0. 246, BIH 1947 and BSHP 2013–2015 $F_{ST}$ = 0.361; $P$ = 0.008).

Bayesian genotype clustering (STRUCTURE) results for samples from all three time periods including the historical BIH samples indicate that $k$ = 3 was the best fit of the data (Figs. 7A and 7B). Assignment to one of the genotype clusters was composed of 1947–1949 individuals from BSHP and BIH only. No individuals from BIH and BSHP historic samples (1947–1949) assigned to the predominant genotype cluster identified for individuals sampled in 2013–2015 and only limited membership was observed in this cluster for individuals from 1988–1991, indicating allelic loss and changes in allele frequency, commensurate with observed declines in allelic richness and private alleles over this same time period (Fig 7C). These results were also supported by the PCoA analysis, which largely separated the 2013–2015 dataset from the 1947–1949 and 1988–1991 datasets on coordinate axis 1 (Fig. 7D). Whereas the BSHP 1948–1949 and 1988–1991 samples showed considerable overlap in PCoA space and the BIH 1947 samples overlapped completely with 1948–1949 BSHP samples.

Although an assignment test approach, such as Bayesian genotype clustering analysis, is not well suited to capture rapidly shifting population dynamics, STRUCTURE results did reveal an increase in population genetic structure among BSHP subpopulations in 2013–2015 compared to 1988–1991. There was no statistical support for population subdivision in the 1988–1991 dataset. However, in the 2013–2015 dataset, there was statistical support for both $k$ = 2 and 4 ($k$ = 2, average LnP(D) = −2492.92, SD ± 1.540, Δk = 33.59457; $k$ = 4, average LnP(D) = −2,466.82, SD ± 9.583, Δ$k$ = 31.943; Fig. S3).

## DISCUSSION

Range-wide levels of genetic diversity and its dispersion across the landscape set the stage for species' adaptive potential under changing climatic conditions. Because maintenance of genetic diversity is often dependent upon movement of individuals within and among local populations (*Booy et al., 2000*; *Bálint et al., 2011*; *Pauls et al., 2013*), it is increasingly important to monitor gene flow among local populations especially in regions of the range that have experienced documented declines. Demographic data alone do not necessarily

Peer J

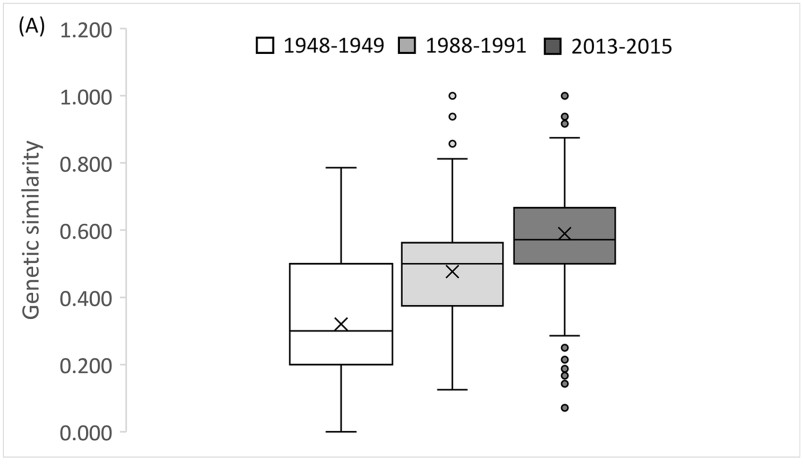

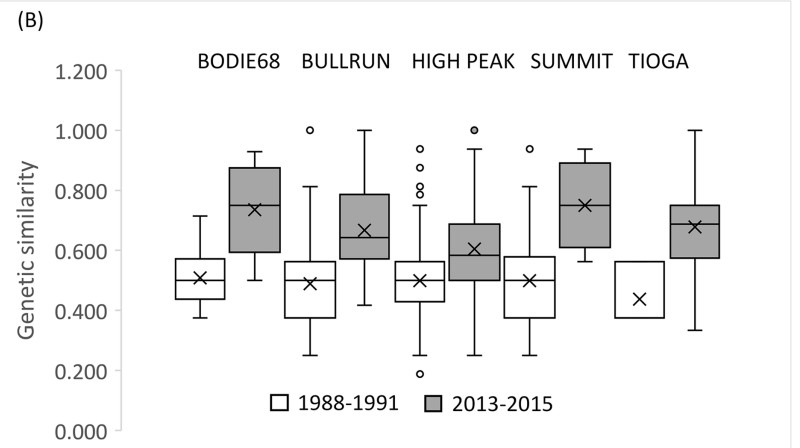

**Figure 6 Genetic similarity comparisons between time periods for entire population and single patch comparisons.** (A) Average genetic similarity among individuals increased in the BSHP population over time (Kruskal-Wallis Test Statistic = 1780.085, df = 2, $P$ = 0.0001). (B) Average genetic similarity among individuals per patch, for patches with sufficient data for comparison ($n \geq 3$ individuals) was significantly higher in 2013–2015 than in 1988-1991 (Mann-Whitney U, 10.5–97,595.5, df = 1, $P$ = 0.000–0.009).          

reflect available genetic resources or long-term trends in patch colonization-extinction dynamics. Therefore, long-term annual census data together with population genetic monitoring provide a robust assessment of patterns of occupancy that influence both long term population viability and evolutionary potential.

Despite past evidence of an active metapopulation dynamic (*Peacock & Smith, 1997*; *Moilanen, Smith & Hanski, 1998*) and significant inter-annual flux in both patch occupancy and dispersion of genetic variation among patches, patch population dynamics at BSHP appear to be changing. The patch occupancy data over the last 51 years together with estimates of genetic diversity provide support for the hypothesis of recent population contraction (*Klingler et al., 2021*) and disrupted metapopulation dynamic (*Peacock & Smith, 1997*). Here we document a reduction in the proportion of patches occupied over time and provide evidence of genetic coalescence and genetic erosion *via* a decline in allelic richness and expected heterozygosity although not observed heterozygosity. Together with
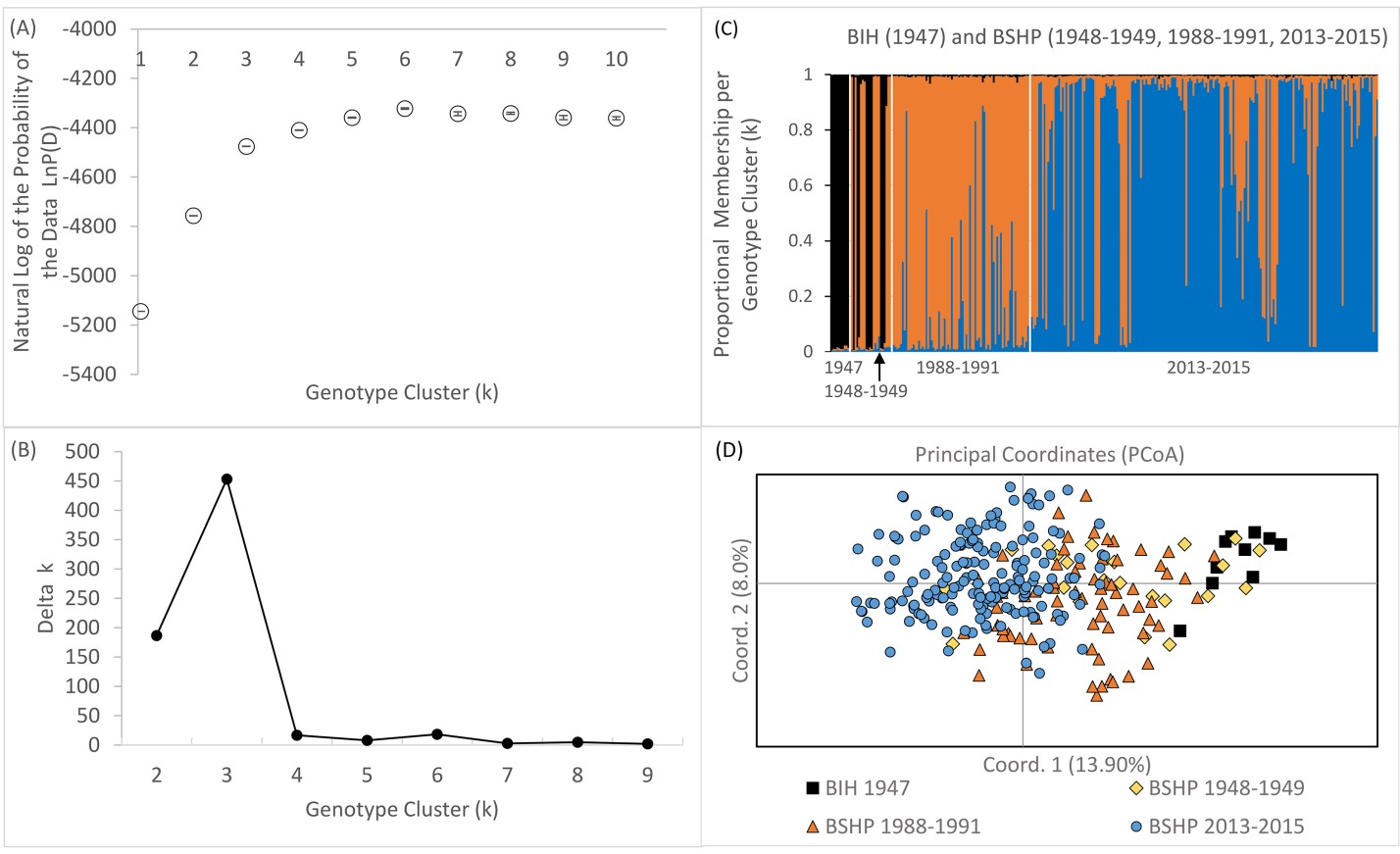

**Figure 7 Bayesian genotype clustering results for BSHP across the three time periods 1948–1949, 1988–1991, 2013–2015 and BIH 1947.** (A) the natural log of the probability of the data LnP(D), (B) delta $k$ values showing $k = 3$ as the best fit of the data, (C) proportional membership per individual per genotype cluster per time period and (D) PCoA results separating BSHP 2013–2015 individuals from 1988–1991, 1948–1949 and BIH 1947 individuals.

a decrease in the effective population size (Ne), these data suggest an increasing isolation among occupied ore dump habitat patches within the BSHP site as well as at the regional scale.

Metapopulation theory predicts accelerated population decline as the probability of re-colonization of extirpated patches and dispersal among extant patches decreases (*Gilpin, 1991*). In addition to the reduction in the proportion of patches occupied at BSHP, we also observed declines in the number of occupants per patch in the northern patch network which has remained consistently occupied over the census period. Both large (four of seven large patches (≥10 territories) and small patches (nine of the 21 patches, 1–4 territories) had decreased occupancy, while 70% (7 of 10) of the medium sized patches (5–9 territories) saw either an increase in territory occupancy or retained the same number of occupants over time. These data suggest changes in patch suitability that may be altering the underlying metapopulation dynamic at this site.

Consistent with a change in the extinction-colonization dynamic the occupied ore dump habitat patches within the BSHP site are becoming more isolated from each other. Genetic variation within the habitat patch network has a larger among-patch partition now

than it did approximately 30 years ago (*Peacock & Smith, 1997*) and there has been a concomitant increase in population genetic structure. In addition, average relatedness (*r*) and genetic similarity at both the population and individual patch level show a shift to higher mean values in 2013–2015 when compared with patch dynamics from 1988–1991. Losses of genetic diversity (as measured by Rs, Pa, and He) and a decline in overall effective population size (Ne) from ~76 individuals during the 1988–1990 time period to ~47 individuals in 2013–2015 are also consistent with a pattern of genetic coalescence as per *Gilpin (1991)*.

Movement dynamics among patches in the earlier *Peacock & Smith (1997)* study, which used both mark-recapture and genetic data to document movement patterns and population genetic structure, suggested a fluid movement dynamic across the northern patch network. Although juvenile pikas tended to settle on the first available territory in the *Peacock & Smith (1997)* study, territory availability varied over space and time such that levels of genetic similarity within and among patch subpopulations was transitory and shifted among years. During 1988–1991, levels of within patch genetic similarity varied widely among years for individuals on both single, relatively isolated patches as well as among individuals inhabiting neighboring groups of patches located on the periphery of the study site (*Peacock & Smith, 1997*). In 2013–2015, however, we observe statistically significant differentiation among not only groups of neighboring peripheral subpopulations but also between large core patches such as High Peak and Tioga Shaft with more peripherally located subpopulations, a new pattern that was not observed in the late 20th century (1988–1991).

Under this scenario of reduction in dispersal and connectivity, losses of genetic variation will accelerate over time *via* random genetic drift–a result of demographic and genetic stochasticity inherent to small populations (*Lande, 1993*; *Reed & Frankham, 2003*; *Whiteley et al., 2015*). However, the fact that BSHP pika metapopulation has retained similar levels of observed heterozygosity (Ho) since the mid-20th century suggests that ongoing juvenile dispersal and movement of alleles among extant patches still occurs despite a potentially altered metapopulation dynamic. For example, it is possible that territorial turnover on natal patches may have become more frequent, leading to increased probability of recruitment to the natal patch for juvenile animals and the likelihood of increased genetic similarity and relatedness within patches as we have observed. Although individuals avoid an increased mortality risk associated with between-patch dispersal by recruiting to their natal patch, decreased rates of among-patch dispersal limits the opportunity for the spread of novel genetic variants and increases loss of genetic variation through random genetic drift. Increased rates of patch turnover, especially on smaller patches, may turn these patches into sink habitats and lead to genetic coalescence of the genetic variation found in the larger, more stable patches.

The BSHP population also increasingly has little to no recolonization potential from natural talus patches within the Bodie Hills region, as these patches are largely extirpated (*Nichols, 2011*). Indeed, the sudden extirpation of the New York Hill pika population (*Nichols, Klingler & Peacock, 2016*), located approximately 20 km from BSHP in sagebrush-scrub and in similar man-made subdivided ore dump habitat, may be the result

of a similar disruption to the metapopulation dynamic and an ensuing extinction vortex. Given these population collapses and increasing isolation across the Bodie Hills region, is unlikely that the BSHP pika metapopulation has experienced significant gene flow with other pika populations in the Bodie Hills since the mid-20th century. This hypothesis of isolation is further supported by the comparison of BSHP with mid-20[th] century historical samples collected at Big Indian Mountain (BIH) in the Wasskuk range, another site in the region about 35 km northeast of BSHP. These mid-century (1947–1949) samples from BSHP and BIH overlap completely in PCA space which suggests that genetic connectivity on the landscape once occurred between the BSHP and BIH populations although likely across generations and *via* intervening habitat patches given that these sites are over 30 km apart from one another and way beyond the dispersal capacity of an individual animal. Loss of landscape level connectivity is further supported by recent survey data showing that most of the remaining natural rock outcroppings in the Bodie Hills are currently unoccupied by pikas despite evidence of recent 20[th] and 21[st] century occupation (*Nichols, 2011*; *Stewart et al., 2015*; *Nichols, Klingler & Peacock, 2016*). With increasing isolation over time the gene pool in BHSP has drifted and now appears completely distinct from the mid-century time period (Fig. 7D).

## CONCLUSIONS

Population level responses to local environmental conditions will ultimately determine species persistence across their ranges. Populations on the edge of their distribution are often considered, at least theoretically, to be less capable of mounting an evolutionary response largely due to limited genetic variation, stressful environments, and smaller population sizes (*Gibson, Van Der Marel & Starzomski, 2009*; *Rogers & Peacock, 2012*; *Bouchard et al., 2017*). Previous research investigating population genetic structure in low elevation sites in the Columbia River Gorge found that pika populations at the edge of their bioclimatic envelope were characterized by relatively low allelic richness and heterozygosity with very little evidence of active gene flow (*Robson, Lamb & Russello, 2016*). The continued loss of genetic diversity through population extirpation or long-term genetic erosion in small, isolated populations may represent a significant evolutionary cost at the species level by reducing its capacity to respond to ongoing environmental change.

This study is the first to characterize temporal patterns of population genetic variation and structure in a single pika population across a sampling period of 51 years for census data and over 50 years for genetic data. Across the last half century, the BHSP population has experienced declines in effective population size and allelic diversity, however the lack of a significant decline in observed heterozygosity suggests that a metapopulation dynamic, albeit altered, may still be a possible buffering mechanism for this population. Indeed, the discrete "patchy" habitat structure at BHSP may have prevented a more severe rate of allelic loss than might have occurred through random genetic drift.

Future projections suggest that in the Sierra Nevada and Great Basin, temperatures will continue to warm significantly especially during the summer season even under low-emission scenarios (+2 °C by 2,100; *Adopted IPCC, 2014*) with significant declines in snowpack (*Cayan et al., 2008*; *Jepsen et al., 2016*). Indeed, over the past 25–30 years,
coincident with warming temperatures, the incidence of extirpations across the American pika's range, primarily in the Great Basin and within the Bodie Hills regions, has been particularly pronounced (*Beever, Brussard & Berger, 2003*; *Stewart et al., 2015*; *Nichols, Klingler & Peacock, 2016*).

Although not directly tested in this study, the declines in patch occupancy at BSHP may be related to warming mean summer temperatures documented for this region over the study period. Such changes may have negatively affected survivorship, recruitment, dispersal and patch occupancy of pikas at BSHP. In fact, several lines of evidence suggest climate change may have played a role in the decline of the BSHP population. The middle and southern portions of the population are dominated by small habitat patches with very few medium or larger sized patches available. The loss of subpopulations on patches in the middle and southern portions of the study site and the lack of recolonization suggests these patches may no longer provide either suitable microclimate and/or access to sufficient forage habitat for pikas. These suboptimal territories likely increase the probability of overwinter mortality, resulting in the collapse of the metapopulation dynamic in this half of the study area. In the near term, with projections suggesting increasing ambient temperatures and changes in precipitation regimes for this region (*Cayan et al., 2008*; *Jepsen et al., 2016*), the smaller ore dumps in the northern half of BSHP may lose their thermal buffering capacity. If these patches no longer provide adequate refugia for pikas, they may start to function as population-sink habitat instead as suggested by the already detected decline in territory occupancy on the small habitat patches over the 51 year census period. This, in turn, will alter the spatial structure and proximity of the remaining talus patches that can be occupied. Indeed, the observed increase in genetic similarity and relatedness within extant patches in the northern half of the study area suggests that some underlying mechanisms related to habitat suitability may already be at work.

Levels of neutral intraspecific and even sub-specific lineage diversity as well as traits associated with life history may differ naturally between pika populations in the range core compared to sites that occur on the trailing edge or range peripheries (*Lesica & Allendorf, 1995*; *Robson, Lamb & Russello, 2016*; *Waterhouse et al., 2017*; *Klingler et al., 2021*). For populations like BHSP, some differences in life history have already been noted including vocalizations (*Conner, 1982*; *Trefry & Hik, 2010*) and litter sizes, which tend to be larger on average in BSHP than those in the Sierra Nevada (3.68 and 3.12 respectively; *Smith, 1978*). Therefore, future research would benefit from analyses of adaptive trait loci for multiple populations across the full range of the American pika, but especially for increasingly isolated populations such as BSHP, which may harbor unique adaptive variants (*Waterhouse et al., 2018*; *Schmidt et al., 2021*). The characterization of adaptive variation for populations across the species range will enable a better understanding of whether isolated populations such as BSHP have enough genetic resources to adapt *in situ*, or whether assisted gene flow from these populations could aid other populations in adapting to climate change (*Wilkening, Ray & Varner, 2015*). However, the rapid decline of effective population size and allelic diversity as observed in this study does not bode well for an adaptation scenario and very likely could result in extirpation of this unique population (*Robson, Lamb & Russello, 2016*; *David & Wright, 2017*).

## ACKNOWLEDGEMENTS

We thank Chris Conroy (UC Berkeley-MVZ) for approval of specimen loans, access to specimens and assistance with sample preparation and shipping. We thank UNR Nevada Genomics Center (http://www.ag.unr.edu/genomics/) for DNA sequencing and fragment analysis. Special thanks to BSHP staff and volunteers for continued logistical support especially California State Park Supervising Rangers, J. Heitzmann, T. Peters, and R. Peek. Field work would not have been possible without K. King and P. Nichols. Thank you to V. Kirchoff and L. Mayfield for laboratory support and assistance.

### Funding

Funds for travel, sample collection, laboratory analysis and genotyping was provided by a Bodie Foundation grant (Bodie Foundation is a 501 (c) (3) non-profit corporation), the University of Nevada (UNR) Ecology, Evolution and Conservation Biology graduate program summer stipend, and the UNR Diana Hadley-Lynch Scholarship awards to Kelly B. Klingler. The funders had no role in study design, data collection and analysis, decision to publish, or preparation of the manuscript.

### Grant Disclosures

The following grant information was disclosed by the authors:
Bodie Foundation.
University of Nevada (UNR) Ecology, Evolution and Conservation Biology Graduate Program Summer Stipend.
UNR Diana Hadley-Lynch Scholarship.

### Competing Interests

The authors declare that they have no competing interests.

### Author Contributions

- Kelly B. Klingler conceived and designed the experiments, performed the experiments, analyzed the data, prepared figures and/or tables, authored or reviewed drafts of the article, and approved the final draft.
- Lyle B. Nichols conceived and designed the experiments, performed the experiments, analyzed the data, prepared figures and/or tables, authored or reviewed drafts of the article, and approved the final draft.
- Evon R. Hekkala performed the experiments, authored or reviewed drafts of the article, and approved the final draft.
- Joseph A. E. Stewart analyzed the data, prepared figures and/or tables, authored or reviewed drafts of the article, and approved the final draft.
- Mary M. Peacock conceived and designed the experiments, performed the experiments, analyzed the data, prepared figures and/or tables, authored or reviewed drafts of the article, and approved the final draft.

## Animal Ethics

The following information was supplied relating to ethical approvals (*i.e.*, approving body and any reference numbers):

Institutional Animal Care and Use Committees (IACUC) at the University of Nevada, Reno.

## Field Study Permissions

The following information was supplied relating to field study approvals (*i.e.*, approving body and any reference numbers):

California Department of Parks and Recreation, California Department of Fish and Wildlife, US Department of Agriculture Forest Service, Inyo National Forest.

## Data Availability

The data is available at Dryad: *Peacock (2023)*, Life on the edge: a changing genetic landscape within an iconic American pika metapopulation over the last half century, Dryad, Dataset, https://doi.org/10.5061/dryad.51c59zwbd.

## Supplemental Information

Supplemental information for this article can be found online at http://dx.doi.org/10.7717/peerj.15962#supplemental-information.

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
