# Peer review of "Life on the edge—a changing genetic landscape within an iconic American pika metapopulation over the last half century"

_PeerJ, doi:10.7717/peerj.15962_

## Round 0.1 · original submission · Minor Revisions

Dear authors,

I am pleased to inform you that both reviewers have expressed their admiration for your interesting and well-written manuscript, highlighting its significant results. However, they have provided some constructive feedback on the analysis and figures, which require minor revisions to improve the manuscript further.

I would like to express my gratitude for your hard work and dedication to this research. Thank you for your efforts in addressing the feedback provided by the reviewers.

Best regards,

Armando Sunny

Reviewer 1 ·

Basic reporting

Overall, most of the basic reporting in the paper is solid: the English is clear and unambiguous, with sufficient field background and context, professional structure, and results relevant to the hypotheses.

Some but not all of the raw data were not shared. While the micros data could be accessed it would be helpful to also have the climate data and the population survey data available through the Dryad link.

Experimental design

No comment. The experimental design is thoughtful and solid.

Validity of the findings

Those data that were provided were robust and statistically sound, but the data that were not provided would be helpful in this assessment (although they are likely confirmatory of many of the paper's findings). Conclusions were well-supported by the data.

Additional comments

This paper was really interesting and an important contribution to our understanding of population change in the Anthropocene.

Reviewer 2 ·

Basic reporting

The authors present a very thorough investigation of the demographic and population genetic history of the Bodie pika population. I am impressed with the amount of work that went into this study! The manuscript is very well written, particularly the introduction, which provides a thorough background and justification for the study.
The figure legends/numbers do not match the in-text citations.

Experimental design

The experimental design is appropriate. However, one major concern is that conducting bottleneck tests on populations with as few as 3 individuals is not appropriate. It would be better instead to calculate for the metapopulation as a whole rather than at the patch level, or only for patches with 20+ genotyped individuals.

Validity of the findings

Overall, the findings are valid and important, with the exception of the bottleneck tests. See comments in sections 2 and 4.

Additional comments

>Rinsing forceps in EtOH does not destroy the DNA, nor remove the potential for cross contamination.
>Fig 4. Were all patches sampled in each time period? If not, could you color them according to the time periods in which they were sampled?
>Figure legends don’t match in-text figure citations
>Figure 7: text is difficult to read. Could combine figures A & B and provide two different regressions lines rather than two separate and mostly redundant figures. Likewise, figures C & D could be combined with number of territories on the x-axis and proportion occupied on the y-axis. In figure D, why does the y-axis go to 1.2 when the maximum proportion of occupied sites is 1?
>Figure 9: Indicate which comparisons are significantly different
>L195: should read Eighty-two? Were all patches surveyed in each year?
>L366: Figure 6 (mean annual precipitation) should be cited here.
>L514: What do you mean by “genetic coalescence”?
>L583-584: There is some overlap in points in the figure 8/11D, so I wouldn’t say they’re “completely distinct”

---

## Round 0.2 · accepted · Accept

Dear Authors,

I am delighted to share the wonderful news that the reviewers have expressed their satisfaction with the corrections made to the manuscript. Consequently, I am thrilled to announce that your work is now ready for publication.

Thank you for considering PeerJ as the platform to publish your fascinating and significant manuscript.

Best regards,

Armando Sunny

Reviewer 1 ·

Basic reporting

The revision has addressed my prior request here.

Experimental design

No comment.

Validity of the findings

No comment.

Additional comments

This is a solid and interesting paper and an important contribution to the field.